# The eGFR Decline as a Risk Factor for Metabolic Syndrome in the Korean General Population: A Longitudinal Study of Individuals with Normal or Mildly Reduced Kidney Function

**DOI:** 10.3390/biomedicines11041102

**Published:** 2023-04-05

**Authors:** Seung Hyun Han, Seung Ku Lee, Chol Shin, Sang Youb Han

**Affiliations:** 1Department of Internal Medicine, Division of Nephrology, Inje University, Ilsan-Paik Hospital, Goyang 10380, Republic of Korea; 2Institute of Human Genomic Study, Korea University Ansan Hospital, Ansan 15355, Republic of Korea; 3Biomedical Research Center, Korea University Ansan Hospital, Ansan 15355, Republic of Korea; 4Division of Pulmonary, Sleep and Critical Care Medicine, College of Medicine, Korea University Ansan Hospital, Ansan 15355, Republic of Korea

**Keywords:** metabolic syndrome, eGFR, general population, kidney

## Abstract

Metabolic syndrome (MS) is a risk factor for the development and progression of chronic kidney disease (CKD). However, it is unclear whether decreased renal function affects MS. Through a longitudinal study, we investigated the effect of estimated glomerular filtration rate (eGFR) changes on MS in participants with an eGFR above 60 mL/min/1.73 m^2^. A cross-sectional (*n* = 7107) and a 14-year longitudinal study (*n* = 3869) were conducted to evaluate the association between MS and eGFR changes from the Korean Genome and Epidemiology Study data. The participants were categorized by their eGFR levels (60–75, 75–90, and 90–105 versus ≥ 105 mL/min/1.73 m^2^). In a cross-sectional analysis, the MS prevalence was significantly increased with a decline in the eGFR in a fully adjusted model. The odds ratio of individuals with an eGFR of 60–75 mL/min/1.73 m^2^ was observed to be the highest (2.894; 95% confidence interval (CI), 1.984–4.223). In the longitudinal analysis, incident MS significantly increased with an eGFR decline in all the models, with the highest hazard ratio in the lowest eGFR group (1.803; 95% CI, 1.286–2.526). In joint interaction analysis, all covariates showed a significant joint effect with an eGFR decline on the incident MS. MS incidents are associated with eGFR changes in the general population without CKD.

## 1. Introduction

Chronic kidney disease (CKD) is a well-known independent risk factor for cardiovascular (CV) morbidity and mortality [1,2,3,4]. The global prevalence of this disease has been noted to be steadily or exponentially increasing over the past few decades. With a current global prevalence estimate of 13.4% (11.7–15.1%), CKD is now a major global concern for both individuals and society [5,6,7]. Furthermore, even a mild reduction in kidney function is reported to be an important risk factor for CV diseases and mortality [8,9,10].

Metabolic syndrome (MS) is a group of metabolic abnormalities that are associated with pathologies such as insulin resistance, hyperinsulinemia, and dyslipidemia. It is defined as the presence of three or more of the following conditions: abdominal obesity (or central obesity), high fasting glucose, low high-density lipoprotein cholesterol (HDL), high triglycerides (TG), and elevated blood pressure (BP) [11]. MS and its components are strongly associated with various non-communicable chronic diseases, including CV disease, type-2 diabetes mellitus (DM), hypertension, and CKD [11,12,13,14]. The prevalence of MS is estimated to be around 20–25% in the adult population worldwide [15,16,17]. Due to the continuous aging of the population and the increasingly unhealthy diet and habits, its consequences continue to increase. This will eventually impose a significant economic burden on both individuals and society [18].

MS is a risk factor for the development and progression of CKD [5,11,19]. There is also evidence that suggests a bidirectional relationship between kidney failure and MS [14,20]. In patients with kidney failure, metabolic abnormalities such as insulin resistance or hyperglycemia were present. Several factors relating to CKD, such as metabolic acidosis, activation of inflammation, and the renin-angiotensin-aldosterone system (RAAS), can induce insulin resistance [21,22,23]. Given that mildly reduced kidney function and MS are both risk factors for CV mortality, elucidating the association between MS and non-CKD patients could support the establishment of an early treatment strategy. However, it is unclear whether renal function plays a role in the development of MS. Furthermore, current research is being conducted on patients with CKD stages 3–5, resulting in a lack of data on the relationship between mildly reduced kidney function and MS. Therefore, in this longitudinal study, we aimed to examine the effect of renal function on MS in the general population with an eGFR above 60 mL/min/1.73 m^2^.

## 2. Materials and Methods

### 2.1. Study Design and Participants

The Korean Genome and Epidemiology Study (KoGES) is a community-based, ongoing cohort study with participants aged 40 to 65 years in Ansan and Ansung, South Korea, since 2001. A detailed study design was described in a previous study [10]. This study was conducted using data from the KoGES from 2005 to 2020. A total of 7515 participants were enrolled and underwent several standardized personal interviews, examinations, and laboratory tests every two years. Of these, we included participants who had normal or mildly reduced kidney function with an eGFR ≥ 60 mL/min/1.73 m^2^. We excluded participants with missing data, CV diseases, or cancers. A total of 7107 participants were evaluated to investigate the association between MS and eGFR decline from cross-sectional data (Figure 1).

For a longitudinal analysis, we excluded participants who had MS at baseline (*n* = 2295) or were lost to follow-up (*n* = 943) during the 14 years. Finally, a total of 3869 participants were included and divided into four groups categorized by their eGFR levels (eGFR ≥ 105, 90 to < 105, 75 to < 90, and 60 to < 75 mL/min/1.73 m^2^). The study was performed in accordance with the Helsinki Declaration of 1975, as revised in 2000, and approved by the institutional Review Board of Korea University Ansan Hospital (2006AS0045).

### 2.2. Definition of Kidney Function and Metabolic Syndrome

Kidney function was assessed using eGFR. The eGFR was calculated using the 2009 Chronic Kidney Disease Epidemiology Collaboration (CKD-EPI) equation [24]. MS was diagnosed based on the diagnostic criteria of the National Cholesterol Education Program Adult Treatment Panel III guidelines with an Asian modification for waist circumference [25]. MS was defined as the clinical status of a participant who had at least three or more of the following conditions: (1) abdominal obesity (waist circumference ≥ 80 cm in women or 90 cm in men), (2) elevated BP (systolic BP ≥ 130 mmHg or diastolic BP ≥ 85 mmHg) or taking antihypertensive medications, (3) high fasting glucose (fasting plasma glucose ≥ 100 mg/dL) or taking glucose level-lowering medicine, (4) hypertriglyceridemia (TG ≥ 150 mg/dL), and (5) low HDL (HDL ≤ 50 mg/dL for women or 40 mg/dL for men). Hypertension was diagnosed when the systolic BP was at least 140 mmHg and/or the diastolic BP was at least 90 mmHg, or the patient was taking antihypertensive medications. DM was defined as a fasting glucose level of at least 126 mg/dL, an HbA1c level of ≥ 6.5%, or patients taking medications for DM.

### 2.3. Assessment of Other Covariates

Participants were surveyed using questionnaires for demographic characteristics (age, sex, and body mass index (BMI)), medical conditions, education, and lifestyle (alcohol consumption, smoking status, and exercise). Women responded to the survey regarding their menopause status. BMI was calculated as body weight (kg) divided by measured height squared (m^2^). The education period is divided into three groups based on the length of academic years: 6 years, 6 to 12 years, and over 12 years. Alcohol consumption (g/day) was measured by quantity, frequency, and type of alcohol. Smoking status was categorized into never, former, and current smokers. We also calculated the amount of exercise in metabolic equivalent of task (MET) per day based on the intensity and duration of exercise.

### 2.4. Statistical Analysis

Data was expressed as means and standard deviations (SD) for continuous variables and as frequencies and percentages for the categorical variables. Testing for linear trends was represented using generalized linear models and Mantel-Haenszel chi-squared tests. We conducted a multivariate logistic regression analysis to estimate the odds ratio (OR) for the presence of MS in relation to eGFR groups with a 95% confidence interval (CI) and *p*-value for the cross-sectional study. To estimate the risk of MS development, we applied Cox proportional hazards regression models. We constructed unadjusted and multivariable-adjusted models for confounding factors. In Model 1, we adjusted for age and sex. In Model 2, we also adjusted for exercise, alcohol consumption, education, smoking, and BMI. Model 3 included menopause for women in addition to Model 2 variables. Joint interaction was calculated using multiplication interaction. A 95% CI was estimated using bootstrap data [26]. For each case, the highest group was used as the reference group. Survival curves for MS development in the eGFR groups were obtained using the Kaplan-Meier estimation method and compared using the log-rank test. A two-tailed *p* < 0.05 was considered statistically significant. All statistical analyses were performed using the SAS statistical software (SAS v.9.4, SAS Institute, Cary, NC, USA).

## 3. Results

### 3.1. Association between Metabolic Syndrome and Kidney Function

#### 3.1.1. Baseline Characteristics

For cross-sectional analyses of the association between MS and eGFR, a total of 7107 participants (mean age, 55.8 ± 8.7 years) were included and classified into four groups based on their degree of eGFR. The baseline characteristics of these groups are summarized in Appendix A. The mean eGFR was 99.2 ± 9.9 mL/min/1.73 m^2^. The prevalence of MS was found in 2295 participants (32.3%) at baseline. Among all the participants, 46.7% had abdominal obesity, 44.4% had elevated BP, 31.5% had hypertriglyceridemia, 13.5% had high fasting glucose levels, and 56.4% had low HDL cholesterol levels. Several factors showed a significant trend according to changes in eGFR. From the highest to the lowest eGFR group, participants showed increasing age, body mass index, waist circumference, lower education, systolic BP, and TG, while alcohol consumption and HDL levels decreased. The prevalence of DM and hypertension also increased as the eGFR decreased.

#### 3.1.2. Prevalent MS and Kidney Function in Logistic Regression Model

We analyzed the association between MS and kidney function in 7107 participants (Appendix A). The prevalence of MS significantly increased from the highest to the lowest eGFR groups in the unadjusted model. This significance also persisted in all three adjusted models, from the sex- and age-adjusted Model 1 to the fully-adjusted Model 3 (all linear *p* trends < 0.001). Surprisingly, even the mildly decreased eGFR group (eGFR of 90–105 mL/min/1.73 m^2^) showed significantly higher odds ratios (ORs) of MS in a fully adjusted model (OR 1.230, 95% CI, 1.054–1.434, *p* = 0.008). As the eGFR decreased, the OR increased for each group in the eGFR 75–90 mL/min/1.73 m^2^ and eGFR 60–75 mL/min/1.73 m^2^ levels (OR 1.775; 95% CI, 1.442–2.185; *p* < 0.001 and OR 2.894; 95% CI, 1.984–4.223, *p* < 0.001), with the highest OR observed in the lowest eGFR group after full adjustment (Appendix A).

We further analyzed the associations in both sexes (Appendix A). Women showed clear trends in both cured and fully-adjusted models. In the fully-adjusted model, women had an increased OR for MS according to the degree of eGFR decline: 1.204 (95% CI, 0.968–1.497), 2.080 (95% CI, 1.557–2.779), and 3.100 (95% CI, 1.785–5.384) compared to the highest eGFR group (*p* for linear trend, 0.037). In men, the ORs were significantly higher in the two lowest eGFR groups (eGFR of 75–90 and 60–75 mL/min/1.73 m^2^) than in the reference eGFR group (*p* for linear trend, < 0.001): 1.453 (95% CI, 1.162–1.817) and 1.683 (95% CI, 1.683–2.600), respectively.

### 3.2. Development of MS According to Kidney Function

#### 3.2.1. Baseline Characteristics

A total of 3869 participants who did not have MS at baseline and were available for the cohort study were included to evaluate the incidence of MS. The baseline characteristics of the patients are shown in Table 1. Of the total study participants, 50.6% were men; the mean age was 54.0 ± 8.1 years, and the mean eGFR was 100.8 ± 9.2 mL/min/1.73 m^2^. Age, smoking status, waist circumference, systolic BP, and TG levels showed significant increasing trends with a decline in eGFR, while the HDL level decreased. However, BMI, exercise, alcohol consumption, fasting glucose levels, and diastolic BP did not show significant trends.

#### 3.2.2. Incident MS and Kidney Function in the Cox Proportional Hazard Model

For incident MS analyses, 3869 participants were included. A total of 2150 (55.6%) new-onset MS events were observed during a median follow-up of 12 years. The incidence of MS was 58.2 (men, 53.0; women, 63.9) per 1000 person-year. All groups of participants with an eGFR < 105 mL/min/1.73 m^2^ showed significantly higher HRs than the reference group (*p* for trend < 0.001), and those with an eGFR range of 60 to 75 mL/min/1.73 m^2^ showed the highest HR (HR 1.994, 95% CI, 1.443–2.775) in the crude model (Table 2). Significant trends still persisted in the adjusted models. After adjustment for age and sex, the HRs showed an increasing tendency from the highest to the lowest eGFR groups (*p* < 0.001). After additional adjustment for education, smoking, exercise, alcohol consumption, and BMI, the trend for HR was significantly sustained.

To exclude the influence of menopause, Model 3 included menopausal women as the adjustment factor. In a fully adjusted model, the group with an eGFR of 90–105 mL/min/1.73 m^2^ showed a significantly higher HR (HR 1.131; CI, 1.011–1.266; *p* = 0.031), and the HRs also showed increasing trends with eGFR ranges of 75 to <90 and 60 to <75 mL/min/1.73 m^2^ (HR 1.273; 95% CI, 1.078–1.502; *p* = 0.004 and HR 1.779; 95% CI, 1.269–2.493; *p* = 0.001, respectively). The four eGFR groups presented significantly different MS-free survival rates using the adjusted Kaplan-Meier curve (Figure 2). The lower eGFR group showed a more rapid decline in MS-free survival during the 14-year follow-up. Interestingly, in the subgroup analysis stratified by sex, the crude and fully-adjusted models showed a significant increase in the development of MS with a declining eGFR in men. However, no significant linear changes were observed in women.

To determine the population at high risk for MS according to renal function, we further analyzed the joint effects of covariates with changes in eGFR on incident MS (Figure 3). All covariates such as age, sex, BMI, exercise, alcohol consumption, smoking status, and education showed a significant joint effect with an eGFR decline on the development of MS. The effects were prominent in participants who were younger, women, ex-smokers, had higher alcohol consumption, lower education, less exercise, and a higher BMI.

## 4. Discussion

The present study indicates that even a small decline in the eGFR range of participants with a normal eGFR range of ≥ 90 mL/min/1.73 m^2^ was a risk factor for the development of MS in the 14-year longitudinal analysis. This risk also shows significant increasing trends as the eGFR declined. Furthermore, all covariates such as age, sex, BMI, exercise, alcohol consumption, smoking status, and education showed a significant joint effect with eGFR decline on the development of MS by interaction analysis. The study also shows that the prevalence of MS is strongly associated with a lower eGFR in the general population without CKD. The prevalence of MS was 32.3% in participants with an eGFR ≥ 60 mL/min/1.73 m^2^. Lower eGFR levels were associated with a higher prevalence of MS.

The association between MS and CKD is clear [5,19]. However, few reports on their association have been published in the general population with an eGFR above 60 mL/min/1.73 m^2^. Hu et al. reported that MS was independently associated with a mildly reduced eGFR (range, 60–90 mL/min/1.73 m^2^) [27]. This cross-sectional study also indicated that obesity, lower HDL levels, and higher TG levels were associated with a mildly reduced eGFR. Yu et al. also reported that MS with obesity was associated with a mildly reduced eGFR of 60–90 mL/min/1.73 m^2^ in men (HR: 1.74, 95% CI: 1.32–2.29) [28]. However, these previous studies had limitations due to their cross-sectional analysis. We also confirmed their association using cross-sectional analysis. Furthermore, the relatively strong causal inference was determined using a 14-year longitudinal cohort study. Even a small change in eGFR could affect the incidence of MS. To the best of our knowledge, this is the first longitudinal cohort study to investigate the relationship between eGFR changes and MS in the general population with an eGFR of 60 mL/min/1.73 m^2^ or higher.

The cumulative incidence of MS varies. Previous reports have reported that it ranges from 6.0% for 8.3 years to 56.1% for 10 years [29,30]. It was 55.6% during the 14 years of our study. The cumulative incidence of MS was higher in women in our study, which is consistent with previous reports [31]. However, the incidence was also noted to be higher in men than in women over a 1-year short-term period [32]. The development of MS with a decline in the eGFR was prominent in men. Previous reports have also shown that men with MS show a more rapid decline in their eGFR than women [33]. The exact mechanism for the difference in the incidence rates of eGFR reduction and MS according to sex is not clear. However, hormonal changes occur during menopause in middle-aged and older women, resulting in metabolic changes such as insulin resistance and increased abdominal adiposity [34]. Reduced estrogen levels in women could serve as a confounding factor in the sex difference in the incidence of MS, with women showing a higher cumulative incidence while men experience a more pronounced increase in MS development with changes in their eGFR. Further studies are needed to clarify these results. This pattern was also significant in participants with a higher BMI. BMI also showed a significant joint effect with changes in eGFR levels based on interaction analysis. Considering that a higher BMI is a well-known risk factor for MS, men or overweight individuals need to be cautious as renal function decreases even for eGFRs > 60 mL/min/1.73 m^2^.

Even a small decrease in kidney function has been reported to increase CV morbidity and mortality [8,35,36]. The Framingham Heart Study reported that individuals with eGFRs of 60–69 and 70–79 mL/min/1.73 m^2^ experienced a higher incidence of CV disease (HR, 1.40; 95% CI, 1.02–1.93 and HR, 1.45; 95% CI, 1.05–2.00, respectively) [8]. Schiffrin et al. reported that patients with an eGFR of 60–90 mL/min/1.73 m^2^ had a 1.5-fold higher OR of CV risk than those with normal kidney function [36]. Currently, the clinical relevance of MS and eGFR in the general population without CKD is not well defined. Our study showed significant associations between MS and eGFR changes. Considering the role of MS and CKD in mediating CV disease and CKD progression [12,37,38,39], our study suggests a possible link between MS, CV disease, and CKD in the general population. Clarification of the relationship between MS and eGFR decline is necessary for establishing early management strategies.

The present study has several strengths. First, it was a community-based cohort study of the general population conducted as a national project that was managed systematically with sufficient statistical power. The questionnaires were standardized, and the follow-up loss was small during the study period. Second, longitudinal analyses could enhance causal inference about the associations between MS and eGFR changes, overcoming the limitations of cross-sectional studies. To the best of our knowledge, this is the first longitudinal cohort study that elucidates the effect of kidney function on MS incidence. Lastly, beyond the CKD condition, we focused on the effect of small eGFR changes on MS in the general population without CKD. These efforts show that a mild eGFR decrease is significantly related to MS. Our results support the need for early strategies to protect patients with mildly reduced eGFRs from MS.

However, this study also had several limitations. First, we investigated the incidence of MS based only on the changes in eGFR. Other signs of kidney damage, such as hematuria, and imaging abnormalities may have affected the results. Further studies are required to clarify the association between MS and other signs of kidney damage. Second, we performed a multivariate regression analysis considering a wide variety of confounding factors; however, other hidden covariates could affect the results. Third, since the eGFR group was divided by the criteria measured at baseline, the decline in eGFR with age followed by the research period was not considered. Therefore, these conditions may have acted as confounding variables. Fourth, because the present study was conducted on middle-aged or older Korean participants, further research on ethnic and racial populations is required.

In conclusion, we confirmed that eGFR decline is associated with an increased risk factor for the incidence of MS in the Korean general population without CKD. These results might have important clinical implications in patients with a mildly reduced eGFR as well as with CKD stages between 3 and 5. Further studies are needed to clarify the relationship between MS and eGFR changes in these individuals.

## Figures and Tables

**Figure 1 biomedicines-11-01102-f001:**
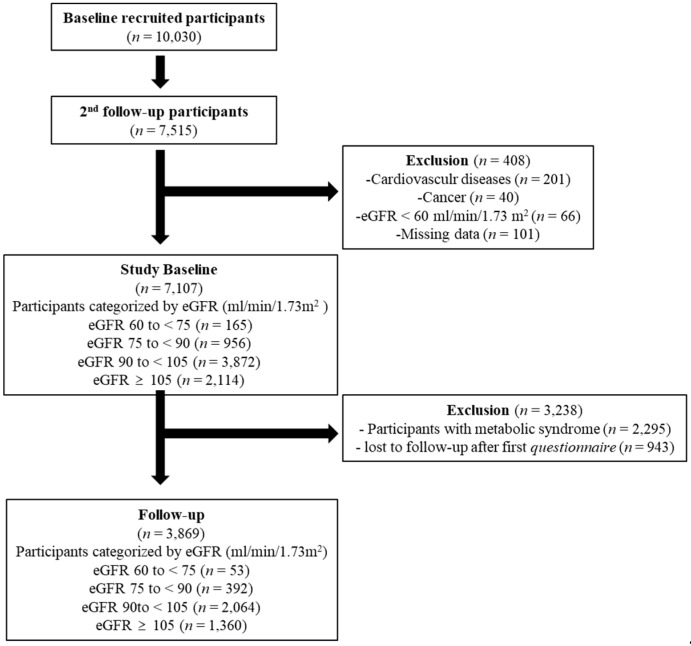
Flowchart of the study participants. eGFR: estimated glomerular filtration rate.

**Figure 2 biomedicines-11-01102-f002:**
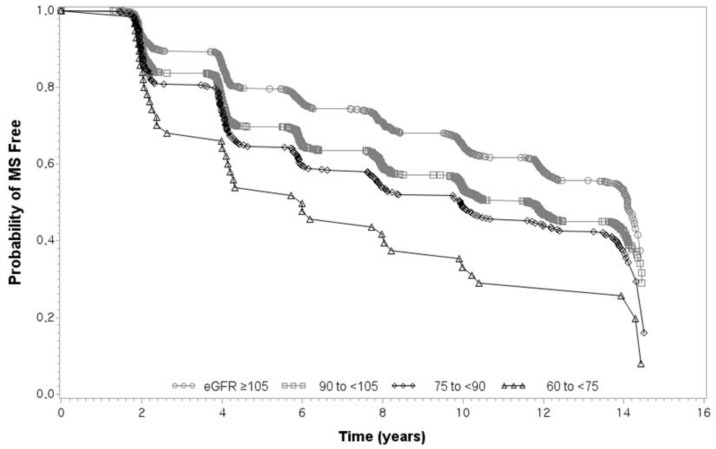
Kaplan–Meier survival curves of incident MS according to eGFR category. Kaplan–Meier survival curves were adjusted for age, sex, education, smoking status, exercise, alcohol consumption, BMI, and menopause for women. MS: metabolic syndrome; eGFR: estimated glomerular filtration rate; BMI: body mass index.

**Figure 3 biomedicines-11-01102-f003:**
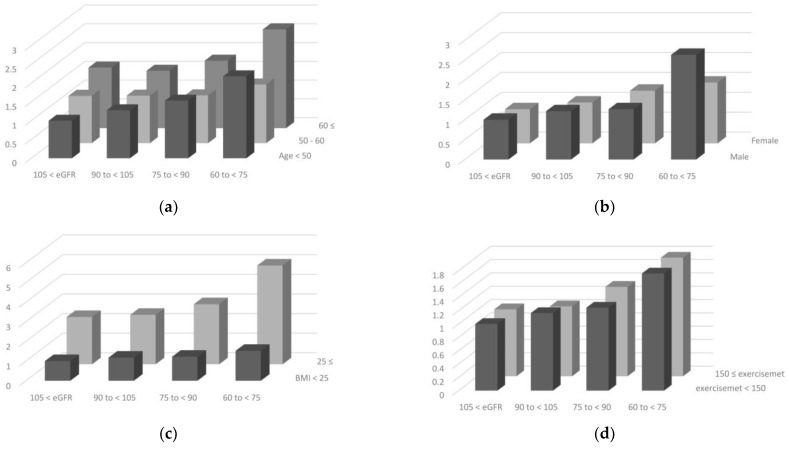
Joint interaction of eGFR with potential confounders on incident metabolic syndrome. Plots show HR and *p*-values for interactions by eGFR category. (**a**) age, (**b**) gender, (**c**) BMI, (**d**) exercise, (**e**) alcohol consumption, (**f**) smoking status, and (**g**) education. HR: hazard ratio; eGFR: estimated glomerular filtration rate; BMI: body mass index; MET: metabolic equivalent of task.

**Table 1 biomedicines-11-01102-t001:** Baseline characteristics of the KoGES cohort study participants based on eGFR category (*n* = 3869).

Characteristics	eGFR Category, mL/min/1.73 m^2^					*p* Value
	Total (*n* = 3869)	eGFR ≥ 105 (*n* = 1360)	eGFR 90 to <105 (*n* = 2064)	eGFR 75 to <90 (*n* = 392)	eGFR 60 to <75 (*n* = 53)	
Age (yr)	54.0 ± 8.1	48.3 ± 4.3	56.3 ± 7.6	60.5 ± 8.9	62.1 ± 7.4	<0.001
Male (%)	1956 (50.6%)	564 (41.5%)	1119 (54.2%)	237 (60.5%)	36 (67.9%)	<0.001
BMI (kg/m^2^)	23.8 ± 2.8	23.7 ± 2.7	23.9 ± 2.8	23.7 ± 2.8	24.3 ± 3.1	0.198
Exercise (MET/d)	129.4 ± 250.9	124.8 ± 303.0	128.4 ± 211.8	146.4 ± 242.4	157.4 ± 241.3	0.277
Alcohol consumption (g/day)	9.3 ± 20.7	10.3 ± 24.4	9.0 ± 18.9	7.3 ± 16.0	8.7 ± 19.6	0.455
Waist circumference (cm)	81.9 ± 8.0	80.1 ± 7.7	82.8 ± 8.0	83.0 ± 7.6	84.1 ± 8.6	<0.001
Smoking status						0.001
Never smoker	2365 (61.1%)	903 (66.4%)	1227 (59.4%)	212 (54.1%)	23 (43.4%)	
Ex-smoker	755 (19.5%)	200 (14.7%)	424 (20.5%)	110 (28.1%)	21 (39.6%)	
Current smoker	749 (19.4%)	257 (18.9%)	413 (20.0%)	70 (17.9%)	9 (17.0%)	
Education						<0.001
<6 years	972 (25.1%)	190 (14.0%)	632 (30.6%)	133 (33.9%)	17 (32.1%)	
6 to <12 years	2272 (58.7%)	927 (68.2%)	1132 (54.8%)	187 (47.7%)	26 (49.1%)	
≥12 years	625 (16.2%)	243 (17.9%)	300 (14.5%)	72 (18.4%)	10 (18.9%)	
Menopuase in women	1135 (29.3%)	244 (17.9%)	742 (35.9%)	132 (33.7%)	17 (32.1%)	<0.001
eGFR (ml/min/1.73 m^2^)	100.8 ± 9.2	109.6 ± 4.2	98.7 ± 4.2	85.2 ± 3.8	69.9 ± 4.4	<0.001
Creatinine (mg/dL)	0.97 ± 0.14	0.88 ± 0.11	0.98 ± 0.12	1.13 ± 0.13	1.31 ± 0.14	<0.001
Urine Protein	13 (1.0%)	26 (1.3%)	8 (2.0%)	2 (3.8%)	13 (1.0%)	0.133
Fasting glucose (mg/dL)	89.8 ± 10.2	89.0 ± 11.0	90.3 ± 9.9	90.7 ± 9.2	88.6 ± 9.2	0.869
SBP (mmHg)	112.4 ± 14.7	108.9 ± 13.3	113.8 ± 15.0	116.1 ± 15.3	116.4 ± 15.6	<0.001
DBP (mmHg)	75.7 ± 9.6	74.4 ± 9.8	76.4 ± 9.5	76.9 ± 9.3	75.6 ± 9.9	0.315
HDL (mg/dL)	46.3 ± 10.2	47.1 ± 10.5	46.1 ± 10.0	45.4 ± 10.4	42.5 ± 9.9	0.001
TG (mg/dL)	113.0 ± 66.7	111.0 ± 76.2	113.1 ± 61.5	116.5 ± 55.0	131.4 ± 72.9	0.022
Abdominal obesity (%)	1160 (30.0%)	351 (25.8%)	669 (32.4%)	125 (31.9%)	15 (28.3%)	0.001
High blood pressure (%)	1041 (26.9%)	258 (19.0%)	602 (29.2%)	155 (39.5%)	26 (49.1%)	<0.001
Hypertriglyceridemia (%)	591 (15.3%)	205 (15.1%)	302 (14.6%)	72 (18.4%)	12 (22.6%)	0.121
High fasting glucose (%)	169 (4.4%)	48 (3.5%)	86 (4.2%)	30 (7.7%)	5 (9.4%)	0.001
Low HDL (%)	1666 (43.1%)	611 (44.9%)	864 (41.9%)	169 (43.1%)	22 (41.5%)	0.362

Data are presented as the mean ± standard deviation. BMI: body mass index; eGFR: estimated glomerular filtration rate; SBP: systolic blood pressure; DBP: diastolic blood pressure; HDL: high-density lipoprotein; TG: triglyceride; MET: metabolic equivalent of task.

**Table 2 biomedicines-11-01102-t002:** Cox proportional hazard model of MS development based on eGFR category in the cohort follow-up period.

eGFR Category, ml/min/1.73 m^2^		HR (95% CI) *p* Value				*p* for Linear Trend
		eGFR ≥ 105 (*n* = 1360)	eGFR 90 to <105 (*n* = 2064)	eGFR 75 to <90 (*n* = 392)	eGFR 60 to <75 (*n* = 53)	
**Total**	Incidence case/total	659/1360	1208/2064	244/392	39/53	
	Person-year	46.6	63.6	71.3	97.8	
	Crude	Reference	1.364 (1.240–1.500) <0.001	1.531 (1.321–1.773) <0.001	1.994 (1.443–2.755) <0.001	<0.001
	Model 1	Reference	1.195 (1.070–1.336) 0.002	1.278 (1.085–1.507) 0.003	1.678 (1.201–2.345) 0.002	<0.001
	Model 2	Reference	1.163 (1.041–1.299) 0.088	1.305 (1.107–1.539) 0.002	1.803 (1.286–2.526) 0.001	<0.001
	Model 3	Reference	1.131 (1.011–1.266) 0.031	1.273 (1.078–1.502) 0.004	1.779 (1.269–2.493) 0.001	<0.001
**Women**	Incidence case/total	393/796	617/945	102/155	15/17	
	Person-year	27.8	32.5	29.8	37.6	
	Crude	Reference	1.593 (1.404–1.809) <0.001	1.681 (1.352–2.091) <0.001	3.644 (2.174–6.107) <0.001	<0.001
	Model 1	Reference	1.087 (0.928–1.273) 0.300	1.025 (0.800–1.312) 0.846	1.813 (1.057–3.110) 0.031	0.274
	Model 2	Reference	1.069 (0.911–1.255) 0.411	1.037 (0.809–1.330) 0.775	1.965 (1.143–3.380) 0.015	0.193
	Model 3	Reference	1.057 (0.899–1.224) 0.501	1.026 (0.799–1.317) 0.843	1.966 (1.142–3.386) 0.015	0.254
**Men**	Incidence case/total	266/564	591/1119	142/237	24/36	
	Person-year	18.8	31.1	41.5	60.2	
	Crude	Reference	1.209 (1.046–1.398) 0.010	1.470 (1.199–1.802) <0.001	1.611 (1.060–2.448) 0.026	<0.001
	Model 1	Reference	1.195 (1.021–1.399) 0.027	1.433 (1.149–1.787) 0.001	1.579 (1.028–2.423) 0.037	0.001
	Model 2	Reference	1.162 (0.992–1.361) 0.063	1.453 (1.162–1.817) 0.001	1.683 (1.090–2.600) 0.019	0.001

Data are presented as a hazard ratio (95% confidence interval) and *p*-value. Person-year was calculated per 1000 person. Model 1: adjusted for age and sex. Model 2: adjusted for age, sex, education, smoking, exercise, alcohol consumption, and BMI. Model 3: adjusted for age, sex, education, smoking, exercise, alcohol consumption, BMI, and menopause for women.

## Data Availability

The data that support the findings of this study are available from the Korea Centers for Disease Control and Prevention, KCDC. Restrictions apply to the availability of these data, which were used under license for this study. Data are available [http://www.cdc.go.kr] with the permission of KCDC.

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
