# Peer review of "The eGFR Decline as a Risk Factor for Metabolic Syndrome in the Korean General Population: A Longitudinal Study of Individuals with Normal or Mildly Reduced Kidney Function"

_biomedicines, 2023, doi:10.3390/biomedicines11041102_

Round 1

Reviewer 1 Report

Dear editor, I have read with interest the article entitled “Decreased eGFR is a risk factor for metabolic syndrome in the general population with normal or slightly reduced renal function: a longitudinal study. By the Authors Seung Hyun Han , Seung Ku Lee , Chol Shin , Sang Youb Han where the authors address the relationship that probably exists between decreased glomerular filtration and the development of metabolic syndrome. Where the authors address the relationship that probably exists between decreased glomerular filtration and the development of metabolic syndrome, in the population of South Korea, the article is interesting, but first I have a series of questions that could help improve Article.

In the title of the article the word “in” appears twice, please correct this, and this could add in the title of the article, that it is in a Korean population, since it is carried out in South Korea.

In addition, the title should be changed since the eGFR decline contributes as a risk factor to the syndrome, not the opposite.

The loss of nephrons decreases with age at a rate of 10 percent per decade. However, the remaining nephrons are capable of adapting to hemodynamic changes and performing adequate filtration, and this does not mean that it is a risk factor in older people. Rather, the pathologies that comprise the metabolic syndrome deteriorate the organs and systems, including the kidney, where glomerular filtration is compromised and if it is also added to the loss in the number of nephrons associated with age, then this is a determining factor. So that in the metabolic syndrome there is a loss of glomerular filtration and this contributes to the syndrome, in a positive feedback process.

Abstract please define CKD, line 15

Line 15-17  please restructure this sentence it's not understood We investigated a longitudinal study to evaluate the effect of estimated glomerular filtration rate (eGFR) changes on MS in participants with eGFR above 60 mL/min/1.73m2 could stay We investigated through a longitudinal study the effect of estimated glomerular …….

Please in the line 20 add “the” before MS

Please in the line 20 add the verb in the past significantly “was” increased

Line 23 please develop CI

Page 1 Introduction section paragraph 2 this section this paragraph is very confusing please can you change it.

Metabolic syndrome is defined as the association between three or more pathologies present in a patient such as hypertension, obesity, insulin resistance, hyperinsulinemia, hypertriglyceridemia, etc... That damage organs and systems such as renal, cardiovascular... It is associated with an increase in an inadequate diet rich in calories and a decrease in physical activity, where the genetic factor is determining, as is gender.

Page 2, line 46 please delete “is well known”, the phrase could stay MS is a risk factor for the development….

Please remove the oration “However, it is not clear……….MS”. This oration is the justification for the investigation and could add after the line 53 “early treatment strategy”.

Please could you make the flowchart bigger it didn't look right?

Page 3, line 81 please develop BP

Page 3, line 86 develop BMI

Page 4 paragraph 2 lines 129-132. Here it is mentioned that the association between both sexes was analyzed, where women showed a tendency. However, in the tables that are presented, the values of women are not found, only that of men, please clarify this

In any of the sections, be it methods, results or discussion, it should be mentioned if the women were in pre or menopause. This is very important because both the metabolic syndrome and the glomerular filtration are affected by the hormonal level. After the menopause, due to the decrease in estrogen levels, the glomerular filtration is compromised and the development of the metabolic syndrome increases. For this reason, the results presented in this article could change.

Also please add in discussion section such as, the estrogens participate in CKD and metabolic syndrome.

Please could you adjust the cells of table 2, so that the distribution of the values remain in a single line and this facilitates the reader.

Please could you increase the long of the figure 2. which is not observe

The same suggestion for figure 3

Page 9 conclusion section please add that it is in a Korean population

Replace reference 24 with the original which is https://doi.org/10.1093/ije/dyv316

If the authors support the results of this article in the KoGES study, and the original study shows values of creatinine, urine pH, urinary protein, and urine volume, in addition to the values of the menopausal state, why not add them in this study and associate them with decreased glomerular filtration and the metabolic syndrome.

Best regards

The reviewer

Reviewer 2 Report

This is an interesting study. Some comments:

1. Authors shuld show in the table 1 the baseline prevalence of each of the components of the MS

1) abdominal obesity (waist circumference ≥ 80 cm in 80
women or 90 cm in men),

(2) elevated BP (systolic BP ≥ 130 mmHg, or diastolic BP ≥ 85 81
mmHg) or taking antihypertensive medications,

(3) high fasting glucose (fasting plasma 82
glucose ≥ 100 mg/dL) or taking glucose level-lowering medicine,

(4) hypertriglyceridemia 83
(TG ≥ 150 mg/dL), and

(5) low HDL (HDL ≤ 50 mg/dL for women, or 40 mg/dL for men).

2. Modes should be adjusted for all 5 components as baseline. Without this adjustment, results are not valid.

3. Authors should explain why they took eGFR vakues >60 and not also lower values (CKD).  In Germany for example, values >60 are considered 'normal'.  CKD starts under 60 here.

Round 2

Reviewer 2 Report

N/A